

# A multi-stage 3D stress field modelling approach exemplified in the Bavarian Molasse Basin

Moritz O. Ziegler[1,2], Oliver Heidbach[2], John Reinecker[3], Anna M. Przybycin[4], and
Magdalena Scheck-Wenderoth[1,5]

[1]Helmholtz Centre Potsdam, German Research Centre for Geosciences, Telegrafenberg, 14473 Potsdam, Germany
[2]University of Potsdam, Institute of Earth and Environmental Science, Karl-Liebknecht-Str. 24-25, 14476 Potsdam, Germany
[3]GeoThermal Engineering GmbH, Baischstrasse 8, 76133 Karlsruhe, Germany
[4]Bundesanstalt für Gewässerkunde, Am Mainzer Tor 1, 56068 Koblenz
[5]RWTH Aachen University, Department of Geology, Geochemistry of Petroleum and Coal, Templergraben 55, 52056 Aachen, Germany

*Correspondence to:* M. O. Ziegler (mziegler@gfz-potsdam.de)

**Abstract.** The knowledge of the contemporary in-situ stress state is a key issue for a safe and sustainable subsurface engineering. However, information on the orientation and magnitudes of the stress state are few and often not available in the areas of interest. Therefore 3D geomechanical numerical modelling is used to estimate the in-situ stress state and the distance of faults from failure for application in subsurface engineering. The main challenge in this approach is to bridge the gap in scales

between the widely scattered data used for calibration of the model and the high resolution in the target area required for the application. We present a multi-stage 3D geomechanical numerical approach which provides a state of the art model of the stress field for a reservoir scale area from widely scattered data records. Therefore we first use a large scale regional model which is calibrated by available stress data and provides the full 3D stress tensor at discrete points in the entire model volume. The modelled stress state is used subsequently for the calibration of a smaller scale model located within the large scale model

in an area without any observed stress data records. We exemplify this approach with two-stages for the area around Munich in the German Molasse Basin. We estimate the scalar values for slip tendency and fracture potential as measures for the criticality of fault reactivation in the reservoir scale model. Furthermore, the modelling results show that variations due to uncertainties in the input data are mainly introduced by the uncertain material properties and missing $S_{Hmax}$ magnitude data. This leads to the conclusion that at this stage the model's reliability depends only on the amount and quality of available input data rather than

on the modelling technique itself. Any improvements of modelling and increases in model reliability can only be achieved by more high-quality data for calibration.

## 1   Introduction

The contemporary in-situ crustal stress field is of key importance for our understanding of geodynamical processes such as natural and induced seismicity (Häring et al., 2008; Gaucher et al., 2015; Scholz, 2002; Heidbach and Ben-Avraham,

2007; Townend and Zoback, 2004; Zang et al., 2014). The stress field is also a critical a priori information for the safe and sustainable engineering of the underground such as wellbore planning and stability, reservoir management, tunnelling, mining,



and underground waste storage (Altmann et al., 2014; Cornet et al., 1997; Fuchs and Müller, 2001; Moeck and Backers, 2011; Tingay et al., 2008; Zang et al., 2013; Ziegler et al., 2015; Zoback, 2010). The quantification of the criticality of the in-situ stress state in terms of fault reactivation in advance of any underground treatment is essential to identify areas of low criticality for a safe and efficient utilisation of the subsurface (Hornbach et al., 2015; Zoback et al., 1985; Häring et al., 2008; Kohl

and Mégel, 2007). In particular the enhancement of permeability through hydraulic fracturing should be achieved without reactivation of sealing faults or inducing seismic events of economic concern (Deichmann and Ernst, 2009; Yoon et al., 2015; Zoback et al., 1985; Townend and Zoback, 2000).

The main focus of current research is to quantify stress changes due to anthropogenic underground usage (McClure and Horne, 2014; Jeanne et al., 2014; Orlecka-Sikora, 2010; Gaucher et al., 2015; Magri et al., 2013). Induced changes of the 3D

stress state in geo-reservoirs are simulated with thermo-hydro-mechanical (THM) models since the treatment of the underground, e.g. the rate of injected fluid or the amount of mass removal is well-known (Kohl and Mégel, 2007; Gaucher et al., 2015; Van Wees et al., 2014; Jeanne et al., 2014; Cacace et al., 2013; Rutqvist et al., 2013; Magri et al., 2013). However, to assess if the subsurface engineering pushes the system into a critical stress state in terms of absolute values, knowledge of the contemporary in-situ stress, i.e. the undisturbed stress state, is essential (Hergert et al., 2015; Häring et al., 2008).

The 3D in-situ stress state can be described with a symmetric tensor of second degree with six independent components (Jaeger et al., 2007; Zang and Stephansson, 2010). Assuming that the vertical stress $S_v$ is one of the principal stresses the number of independent unknowns reduces to four (Zoback, 2010). In the principal axis system these are the orientation of one of the two principal horizontal stresses, i.e. the maximum and minimum horizontal stress $S_{Hmax}$ and $S_{hmin}$, respectively, as well as the magnitudes of $S_v$, $S_{Hmax}$ and $S_{hmin}$ (Zoback, 2010; Schmitt et al., 2012). Thus, the orientation of this so-called reduced

stress tensor is described by the $S_{Hmax}$ orientation which is systematically compiled by the World Stress Map (WSM) project (Heidbach et al., 2010, 2008; Sperner et al., 2003; Zoback, 1992).

Fig. 1 shows a stress map with a typical density of $S_{Hmax}$ orientation data records with 172 reliable data records for the $82,000 \text{km}^2$ large part of the Alpine Foreland Molasse (Reiter et al., 2015; Reinecker et al., 2010; Heidbach and Reinecker, 2013). This results in an average data density of 0.21 data records per $100 \text{km}^2$ which is the typical claim size for exploration.

In general, the orientation of the stress field does not change with depth in the upper crust (Rajabi et al., 2016; Pierdominici and Heidbach, 2012; Heidbach et al., 2007). Laterally the stress field in the Alpine Molasse rotates only gently counter-clockwise from East to West (Reinecker et al., 2010). Thus, the available stress orientation data allows the determination of the orientation of the reduced stress tensor to a relatively high degree of reliability (Heidbach et al., 2007; Ziegler et al., 2016; Reiter et al., 2015).

More important for the assessment of the criticality is the estimation of the differential stress between the magnitudes of the largest and smallest principal stresses and their changes during stimulation and production. The $S_v$ magnitude can be derived from the vertical density distribution. In contrast to this the horizontal stress magnitudes originate from the geologic history and ongoing tectonic evolution and cannot be determined directly from rock properties (Brown and Hoek, 1978; Zang et al., 2012; Zang and Stephansson, 2010). Furthermore, the increase of horizontal stress magnitudes with depth is often described

with a linear gradient, which is only justified when rock strength and density do not change significantly with depth (Brudy



et al., 1997; Lund and Zoback, 1999). In sedimentary basins this linear increase cannot always be assumed. Competent layers e.g. from the Malm and Muschelkalk alternate with weaker layers with high clay content such as Dogger and Keuper and result in a sudden deviation of the stresses from a linear trend across these layers (Warpinski, 1989; Hergert et al., 2015; Cornet and Röckel, 2012; Gunzburger and Cornet, 2007; Zang et al., 2012). Moreover, the density of stress magnitude data records is in general up to two magnitudes lower than the one of the orientation data (Fig. 1).

To summarize, our knowledge of the 3D in-situ stress state is based on sparsely distributed and incomplete information. In particular the crustal in-situ stress magnitudes are under-determined since they vary laterally and vertically. To determine the full stress tensor for every point in a volume a 3D geomechanical-numerical model workflow is essential that uses the available stress information as model-independent constraints for calibration. Moreover, at reservoir scale often no stress information is available for model calibration (Fig. 1). Thus it is necessary to enlarge the model area until sufficient stress data are within the model volume. In the Bavarian Molasse Basin which we use as an example this enlargement of the model area leads to an increase in model size from a $10 \times 10 \mathrm{km}^2$ reservoir sized model to $70 \times 70 \mathrm{km}^2$ regional model (Fig. 1). Considering a constant resolution this enlargement would lead to a higher number of model degrees of freedom by a factor of 50. In most cases of THM reservoir modelling this is beyond feasibility due to the time required for iterations and limitations in computation power. One option to avoid a high number in degrees of freedom is to refine the structure only in the area of interest (Jeanne et al., 2014; Westerhaus et al., 2008) (Fig. 2a). However, this becomes challenging when local structures have to be integrated. An alternative option is to use nested modelling which is applied in various scientific disciplines such as meteorology, climate simulations, and the simulation of seismic cycles (Warner and Hsu, 2000; Cacas et al., 2001; Giorgi et al., 1998; Hergert and Heidbach, 2011). Essentially a nested modelling approach can be (1) a local high resolution grid inside a coarse grid where the variables are matched at the boundaries (Fig. 2b)(Oey and Chen, 1992), or (2) a multi-stage approach of two or more individual models which increase their resolution within the same area or a subarea (Warner and Hsu, 2000) (Fig. 2c). In contrast to the previously named nested approaches the multi-stage procedure is most favourable in terms of required workload (fast and simple mesh generation) and quality of results (high spatial resolution in the area of interest). Furthermore, it may serve several individual reservoir model locations within the regional area model volume (Fig. 2c). However, so far this procedure has not been applied in 3D geomechanical-numerical modelling of the crustal stress field.

In this paper we demonstrate the applicability of the multi-stage nesting workflow for the 3D geomechanical modelling of the stress tensor. We exemplify our approach with a 3D model of the Greater Munich area in the Northern Alpine Molasse Basin and a generic reservoir model (Fig. 1). We demonstrate the conceptual advantages of the multi-stage approach as a detailed and yet fast workflow for exploration from planning to exploitation. Furthermore, we quantify the impact of the uncertainties of the model parameters and the limited amount of calibration data on the model results and discuss the reliability of the 3D geomechanical numerical modelling.



## 2   Geological setting

The Northern Alpine Molasse Basin is a typical asymmetric foreland basin which extends over $1,000$km from Lake Geneva in the West to Lower Austria in the East (Bachmann et al., 1987). Its widest N-S extent is with 130km in southern Germany (Lemcke, 1988). The basin consists of mainly Tertiary sediments on top of Mesozoic successions and a Variscian basement with

Permo-Carboniferous troughs (Lemcke, 1988; Bachmann et al., 1987). In the Foreland Molasse these sediments dip towards the South where a maximum thickness of approximately $6,000$m is reached in front of and beneath the Alpine mountain chain and the Folded Molasse (Fig. 1) (Bachmann et al., 1987). Due to the Molasse Basin's close link to the Alpine orogeny (Schmid et al., 2008) most of the main faults in the Bavarian Foreland Molasse are steeply dipping (>60°) and strike at least sub-parallel to the Alpine front approximately E-W (Reinecker et al., 2010; Bachmann et al., 1987; Lemcke, 1988). They are considered

mostly inactive at the moment (Reinecker et al., 2010; Bachmann et al., 1987; Lemcke, 1988).

For our model geometry we use the 3D structural model of the Northern Alpine Foreland Basin by Przybycin (2015) which covers the entire German part of the Molasse Basin. It provides 12 stratigraphical units in total with a focus on the Malm and Purbeck, two target horizons for geothermal exploration (Lemcke, 1988; Bachmann et al., 1982; Fritzer et al., 2012). The lateral resolution of the structural model ($1 \times 1.7$km$^2$) is sufficient to provide the geometry for our regional scale model of the

Greater Munich area. The structure is based on freely available data on the depth and thicknesses of stratigraphic units from wells and seismic lines as well as 3D gravity modelling as a further constraint (Przybycin, 2015). The part of the structural model used for the geomechanical model has a size of $70 \times 70$km$^2$ and is referred to as the *root* model. The generic reservoir model located within the root model volume is called *branch* model. It has a size of $10 \times 10$km$^2$ with more detailed structural information e.g. provided by a 3D seismic survey.

## 3   In-situ stress data

### 3.1   Orientation of $S_{Hmax}$

Within the root model area (Fig. 1, orange box) 18 reliable $S_{Hmax}$ orientation data records are located, while there are none in the branch model area (Fig. 1, black box). These data are exclusively from borehole measurements using drilling induced tensile fractures (Aadnoy, 1990) and borehole breakouts (Bell and Gough, 1979; Bell, 1996) as indicators for the $S_{Hmax}$ orientation

(Reinecker et al., 2010). In 15 wells in the model area borehole breakouts are found which have a combined length of 7.7km. In 3 wells drilling induced fractures are found which have a combined length of 0.3km. The stress indicators are found mainly between the surface and a depth of $2-3$km even though some are located at greater depth (Fig. 1). No significant stress rotation or perturbation with depth is observed in the available data (Reinecker et al., 2010). The quality of the data is exceptionally good according to the WSM quality ranking (Heidbach et al., 2010; Sperner et al., 2003; Zoback, 1992) with 8 A-quality

data records (i.e. an uncertainty of ±15°), 6 B-quality data records (±15-20°), and 4 C-quality data records (±20-25°). Under the assumption that $S_v$ is a principal stress component the reduced 3D stress tensor within the model area has a mean $S_{Hmax}$ orientation of $1.7° \pm 19.2°$ which is approximately perpendicular to the Alpine front (Fig. 1).





## 3.2 Stress magnitudes

The magnitude of $S_v$ can be estimated with a relatively high reliability from the thickness of the different overlying units ($z$) in the structural model provided by Przybycin (2015), the density of the corresponding rock material ($\rho_{rock}$, Table 1) and the gravitational acceleration ($g$) given by

$$S_v = \sigma_{zz} = \rho_{rock}gz \tag{1}$$

However, information on the horizontal stress magnitudes is sparse even within the root model area. The magnitude of $S_{hmin}$ is usually derived from hydraulic fracturing (Haimson and Fairhurst, 1969; Hubbert and Willis, 1972), but such data is not available publicly for the Bavarian Molasse Basin. Alternatively, leak-off tests (LOTs), which rely on a cheaper and faster method, are more frequently used for the estimation of $S_{hmin}$. They provide information on the break-down pressure of the tested formation which is then used as an approximation for the $S_{hmin}$ magnitude (Haimson and Fairhurst, 1969; Bell, 1990; Zang et al., 2012). Further information on the $S_{hmin}$ magnitude can be derived from a formation integrity test (FIT). It does not fracture the rock but provides a minimum pressure value at which the rock is stable which in turn provides a lower bound for the $S_{hmin}$ magnitude (Zoback et al., 2003). Even though no hydraulic fracturing was done in the model area a LOT has been conducted in the Unterhaching Gt 1/1a borehole which is used for calibration (T. Fritzer, pers. comm.). Furthermore, several FITs have been performed in the borehole Sauerlach (Fig. 1) that is in the root model area (Seithel et al., 2015). In contrast to the LOTs FITs are not used for calibration since the difference between the FIT pressure and the actual magnitude of $S_{hmin}$ is not known. However, during one of the FITs in the Sauerlach borehole bore fluid was lost into the formation (T. Fritzer, pers. comm.). Hence a leak-off occurred and this FIT is treated as a LOT and also used for the model calibration.

The direct estimation of the $S_{Hmax}$ magnitude would only be possible with overcoring measurements (Hast, 1969; Sjöberg et al., 2003). In addition, reasonable values for the $S_{Hmax}$ magnitude can be derived on the basis of the frictional equilibrium theory (Zoback et al., 2003) or physics based relations whose reliability is largely dependent on the quality of the $S_{hmin}$ magnitude estimation (Zoback, 2010; Cornet, 2015). Seithel et al. (2015) use the friction equilibrium approach and derive a single $S_{Hmax}$ magnitude between 112 and 116MPa at a depth of 4km. We use an $S_{Hmax}$ magnitude of 112MPa in Sauerlach for the calibration even though the uncertainties introduced by the derivation are high. The impact of these high uncertainties on the model results is discussed later on.

## 3.3 Stress regime

In areas with a low number of magnitude stress data records the stress regime provides information on the relative magnitudes of $S_v$, $S_{Hmax}$, and $S_{hmin}$. The stress regime is mainly derived from focal mechanisms of seismic events and to a small extent from geological indicators or hydraulic fracturing experiments (Zoback, 1992; Sperner et al., 2003). In the Swiss part of the Northern Alpine Molasse Basin mainly a strike slip ($S_{Hmax} > S_v > S_{hmin}$) and to a smaller extent extensional ($S_v > S_{Hmax} > S_{hmin}$) stress regime is observed (Heidbach and Reinecker, 2013). However, in the Bavarian Molasse Basin north of the Alpine



front no natural seismicity has been recorded (Grünthal, 2011; Grünthal and Wahlström, 2012) to derive the stress regime from focal mechanisms.

Information from structural geology observing steeply dipping faults in the Bavarian Molasse Basin (Bachmann et al., 1987; Lemcke, 1988) indicates an extensional tectonic faulting regime (Anderson, 1905, 1951). In contrast to this Illies and Greiner (1978); Lemcke (1988), and Reinecker et al. (2010) propose a compressional ($S_{Hmax} > S_{hmin} > S_v$) or strike-slip stress regime in the Molasse Basin. Also Seithel et al. (2015) propose a strike-slip stress regime at a depth of 4km for the Sauerlach project according to their analysis based on the frictional equilibrium theory. However, without further estimations of the stress magnitudes in other depth sections and locations the regional tectonic stress regime setting is subject to large uncertainties.

## 4   Model workflow

### 4.1   Model setup

Both the regional scale root model and the reservoir scale branch model are based on the same modelling assumptions. Assuming that accelerations other than gravity can be neglected the models solve the partial differential equation of the equilibrium of forces. Furthermore, we assume a linear elastic rheology and solve for absolute stresses (no pore pressure). The general model procedure follows the technical workflow explained in detail by Hergert et al. (2015) and Reiter and Heidbach (2014).

The root model extends $70 \times 70 \times 10 \text{km}^3$ in the entire Greater Munich area (Fig. 1). It consists of six different stratigraphic layers (Table 1) based on the 3D structural model by Przybycin (2015). The generic branch model of a potential geothermal site has a size of $10 \times 10 \times 10 \text{km}^3$ and includes six different stratigraphic units (Table 1). For both models the boundaries are oriented perpendicular and parallel to the orientation of $S_{Hmax}$ and $S_{hmin}$ respectively (Fig. 1). Both models are populated with the Young's modulus, the Poisson ratio and the density according to the stratigraphic units (Table 1).

An exact fit of the overburden $S_v$ is achieved by the application of gravity provided that the density of the stratigraphic units is correctly chosen. We implement an equilibrated initial stress state close to lithostatic ($S_{Hmax} \approx S_{hmin} \approx S_v$). Dirichlet boundary conditions (i.e. displacement) are applied to the sidewalls of the model to create horizontal differential stresses that originate from the geologic history and tectonic evolution. The boundary conditions are adjusted to fit the observed magnitudes of $S_{Hmax}$ and $S_{hmin}$ at the calibration points.

Due to the complex topology of the stratigraphy and inhomogeneous rock properties of the different units the finite element method (FEM) that allows unstructured meshes is used to solve the partial differential equation of the equilibrium of forces at discrete points. Thus, both models are discretised into finite elements meshes. The root model is constructed with approximately $10^6$ hexahedral elements resulting in approximately 400m of horizontal and between 15m and 700m of vertical resolution (Fig. 3). A vertically refined resolution is created in the units of interest for geothermal exploration, namely the Malm and Purbeck formation. The Cretaceous and the Triassic (Pre Malm) are only preserved in parts of the root model. Compared to the root model a significantly finer resolution with a total of $21 \times 10^6$ tetrahedral elements is chosen in the branch model. The edge length of the elements varies between 10m and 160m with the coarsest resolution located at the bottom and the edges of the model and the highest resolution in the Purbeck and Malm units of interest for geothermal exploration (Fig. 3).



## 4.2   Model calibration procedure and two-stage approach

The calibration of the root model with stress magnitude data is achieved by the application of two Dirichlet boundary conditions each on one of the perpendicular sidewalls of the model (Fig. 4, left row). A single $S_{hmin}$ magnitude data record can be exactly modelled by a certain combination of two boundary conditions. More precisely an unlimited combination of two boundary

conditions exist to achieve an exact fit of a single $S_{hmin}$ magnitude calibration point. This unlimited amount of combinations of displacement boundary conditions is a linear function of the E-W and N-S displacements and is displayed as a linear slope in Fig. 4a with displacement in N-S direction on the x-axis and displacement in E-W direction on the y-axis. Due to the assumed linear elastic model rheology each combination of East-West and North-South displacement which lies on the slope leads to an exactly calibrated model (Fig. 4a).

If several $S_{hmin}$ magnitudes are available for calibration each of them individually can be exactly reproduced by an unlimited amount of combinations of displacement boundary conditions. However, to achieve a calibration which works for all of the observed $S_{hmin}$ magnitudes a single "best-fit" slope is derived from the linear slopes for the individual calibration points by a linear regression (Fig. 4b)(Reiter and Heidbach, 2014). Each combination of displacement boundary conditions specified by this slope results in a "best-fit" model for all of the calibration points considered.

The same procedure is applied for the calibration of $S_{Hmax}$ magnitudes so that eventually a "best-fit" slope for both the $S_{Hmax}$ and $S_{hmin}$ magnitude stress data records used for calibration are available (Fig. 4c). Displacement boundary conditions defined by the point where these two "best-fit" slopes intersect are used to compute the "best-fit" model that reproduces the $S_{Hmax}$ and $S_{hmin}$ stress data records best (Fig. 4c)(Reiter and Heidbach, 2014).

Application of this calibration procedure is fast and simple since the "best-fit" boundary conditions can be found by the

combination of two linear slopes based on the calibration data and the displacement boundary conditions. Therefore, to find the "best-fit" boundary conditions only three different models with arbitrary displacement boundary conditions are required (Fig. 5a). The modelled $S_{Hmax}$ and $S_{hmin}$ magnitudes at the location of calibration points in each of the three models are compared to the actually observed data records (Fig. 5b,c). A linear regression with two unknown variables leads to the "best-fit" slopes for the combination of boundary conditions to model the $S_{Hmax}$ and $S_{hmin}$ magnitude, respectively (Fig. 5d). At the

intersection of the two slopes the boundary conditions for the "best-fit" model can be derived (Fig. 5d).

It is assumed that the stress data records used for the calibration are the results of the far-field stress state and its interaction with structural features such as the local density and/or the strength contrasts represented within the root model. If the measurements were e.g. the result of an unknown or not implemented local active fault the results of the calibration would not be reliable. Thus, in general the data used for calibration should be representative for a large volume of the individual lithological

layer.

Under this assumption the "best-fit" model simulates the stress state at discrete points in the entire model volume. Hence, information on the stress state is now available also in areas of the root model where previously no observables on the stress state were available. This means that also in the branch model which does not include any observed stress data records now simulated information on the stress state are available from the root model to calibrate the branch model (Fig.5d,f).




Since the calibration of the branch model is done in the same way as for the root model but rather with a simulated stress state from the root model as calibration points than observed stress data records a large number of potential calibration points with $S_{hmin}$ and $S_{Hmax}$ magnitudes are available. The $S_{Hmax}$ and $S_{hmin}$ magnitudes at each calibration point can be modelled individually in the branch model by combinations of boundary conditions, each described by a linear slope (Fig. 4a). For all $S_{Hmax}$ and $S_{hmin}$ magnitudes, respectively, a best-fit slope which is based on the individual linear slopes is derived (Fig. 4b). Two best-fit slopes describe the combinations of boundary conditions which model the $S_{Hmax}$ and $S_{hmin}$ magnitudes, respectively, best. The intersection of these two best-fit slopes defines the boundary conditions which are used to compute the "best-fit" branch model (Fig. 4c). This calibration procedure is performed analogously to that of the root model (Fig. 5e-h).

For a successful transfer of the stress state from the root to the branch model it is critical that the stress state used for the calibration of the branch model is obtained at discrete points of the root model and not in its volume. Otherwise the stress state extracted from the root model is potentially biased due to interpolations from discrete points into the volume which are performed by the visualisation software. Since the large amount of possible calibration points allows choosing them arbitrarily, their location needs to be considered. We recommend using calibration points close to the border of the branch model but outside the zone prone to boundary effects. Calibration points from the root model in the centre of the branch model are a contradiction of the two-stage approach which aims at finding local stress changes due to high resolution structural features that are only present in the branch model. Due to the lack of any other stress data in the branch model area the calibration procedure imposes the root model's basic stress state on the branch model which prevents such local stress perturbations. Hence this necessary imposition should be reduced to the boundaries of the branch model that are not used for interpretation anyway. Furthermore, the calibration points should be evenly distributed along the branch model boundary and also represented in all stratigraphic units to account for different material properties. Special attention needs to be paid to units which are either only present in the root or the branch model or have a significantly different geometry or rock properties in the two models.

## 5 Model results

In the following two sections we present the results of two model scenarios for the root model that fit equally well the observed stress data, but with different stress regimes (Fig. 6). For the branch model we present our results on one scenario that can be considered as our "best-fit" model (Fig. 7).

### 5.1 Root model

The "best-fit" root model of the stress state at discrete points in the Greater Munich area is calibrated using $S_{hmin}$ magnitudes from the two LOTs and one $S_{Hmax}$ magnitude described in detail in the stress data section 3.2. The "best-fit" model has a good fit to the three $S_{hmin}$ calibration data points and an almost perfect fit for the single $S_{Hmax}$ calibration point. Deviations between observed and modelled data are on average 0.4MPa for the two $S_{hmin}$ calibration points and 0.04MPa for the single $S_{Hmax}$ calibration point.





Fig. 8 shows the "best-fit" model results along the Sauerlach borehole profile along with the FIT data of Seithel et al. (2015). The black line shows the vertical stress magnitude with depth which depends only on the chosen rock density. The blue line is the $S_{hmin}$ magnitude which is larger than all FIT values at all depth sections. The blue star represents the magnitude and depth of the $S_{hmin}$ magnitude inferred from a FIT with leak-off. The red line is the $S_{Hmax}$ magnitude in the "best-fit" model while the

dashed line represents $S_{Hmax}$ in another model scenario. The red star marks the depth and magnitude of $S_{Hmax}$ in the "best-fit" model. The shaded areas show the modelled magnitudes for model scenarios which use $S_{Hmax}$ magnitudes between 92MPa and 118MPa in a depth of 4km below the Sauerlach site for calibration. This demonstrates that the single $S_{Hmax}$ magnitude derived in conjunction with the ambiguity of the stress regime opens up a wide range of model scenarios which all equally well fit the $S_{hmin}$ data. Even though a compressional regime can be excluded by the available data in Sauerlach no indications exist

whether $S_{Hmax} > S_v$ or $S_{Hmax} < S_v$. That means that the prevalence of a normal faulting or a strike-slip stress regime is possible. To account for this variability several different scenarios have been computed of which two are compared in Fig. 6. Note that the only difference between these scenarios is the $S_{Hmax}$ magnitude value used for the root model calibration (Fig. 6a 96MPa, Fig 6b 112MPa); the fit to the $S_{hmin}$ data from the LOTs is equally good (Fig. 6).

In Fig. 6 we show for the aforementioned two model scenarios a number of scalar stress values derived from the modelled

3D stress tensor on cross sections and within stratigraphic units. The figure shows that the values vary depending on the stratigraphic units horizontally and laterally. More important, the results from the two model scenarios fitting the model-independent calibration data equally well are quite different. The first row of Fig. 6 shows the variability of the stress regime using a continuous scale, the so-called regime stress ratio (RSR) from Simpson (1997). Close to the surface a strike slip regime dominates with compressional components in some areas. With increasing depths this changes to a prevailing extensional

regime. Moreover, some changes from strike slip to extensional and back to strike slip can be observed which are not a smooth linear trend but highly dependent on the lithology.

The second row of Fig. 6 displays the horizontal stress anisotropy as a stress magnitude ratio of $S_{Hmax}/S_{hmin}$ on a N-S and E-W cross section through the two model scenarios of the root model. It is clearly visible that the ratio varies significantly with depth and between the model scenarios. The southward dipping Malm and Purbeck units have stress ratios of up to 0.15 higher

than the basement layer and overlying sediments, respectively.

The last row in Fig. 6 shows the differential stress in the middle of the Malm unit. Both model scenarios show higher differential stresses in the South where the Malm units are deeper than in the North. The largest N-S difference is 7MPa in model scenario a) in contrast to 12MPa in model scenario b), even though the relative pattern of the differential stresses in the Malm unit is quite similar in both model scenarios. This pattern highlights the main trend of an increasing differential stress

towards the South. At the same time significant changes of the differential stress within less than 10km of up to 1MPa are predicted.

## 5.2 Branch model

In this section we show the model results of the branch model (Fig. 7) that uses the stress data derived from the root model scenario displayed in the right row of Fig. 6. In order to visualise the criticality of the reservoir we use two scalar values which





are computed from the modelled 3D stress state. The first one is the fracture potential (FP) of intact rock volume (Connolly and Cosgrove, 1999). It is computed as

$$FP = \frac{\text{actual maximum shear stress}}{\text{acceptable shear stress}} \tag{2}$$

$$= \frac{0.5(S1 - S3)}{C\cos(\Phi) + 0.5(S1 + S3)\sin(\Phi)} \tag{3}$$

with S1 and S3 as the maximum and minimum principal stress, C as the cohesion, and $\Phi$ as the friction angle. As a second scalar value slip tendency (ST) is computed on faults (Morris et al., 1996). It is a measure of the criticality of faults which is illustrated as a scalar value for the distance to failure derived from the stress tensor with values between 0 (safe) and 1 (failure). Slip tendency is computed for faults or fault segments of a certain orientation and is defined as

$$ST = \frac{\frac{\tau_{max}C}{\sigma_n}}{\mu} = \frac{\frac{\tau_{max}C}{\sigma_n}}{\tan(\Phi)} \tag{4}$$

with the maximum shear stress $\tau_{max}$, the normal stress $\sigma_n$, the friction angle $\Phi = \arctan(\mu)$, and the friction coefficient $\mu$. The application of these two values is shown in the branch model with generic faults in Fig. 7.

    The high dependence of slip tendency on the orientation, friction, and cohesion of the fault is displayed in Fig. 7. A high variability of slip tendency between 0.05 and 0.3 is observed on the generic faults. This variability is induced by the 3D stress tensor and the curved fault surfaces. Furthermore, due to differently assumed friction and cohesion of the rocks the Malm $\delta$ −

Purbeck units have a clearly smaller value of slip tendency compared to the Chattian units in the hanging wall and the Malm $\alpha - \gamma$ in the footwall. The fracture potential in the basement lies between 0.1 and 0.2 which is quite low and hence requires high pressure for hydraulic fracturing operations to enhance the permeability of the fracture network.

    Information provided by the branch model is used in an early pre-drilling stage of a project to assess if the initial conditions of the reservoir and its criticality allow a safe production, i.e. both slip tendency and fracture potential have low values as in

Fig. 7. Before the drilling of the borehole begins the planning of the drillpaths can be optimised. Especially if intersections with faults are required their paths can be monitored and adapted in a way that they circumnavigate fault segments which have a higher value of slip tendency which means that this fault segment is more favourably oriented for a potential failure compared to other fault segments. In Fig. 7 areas with cool colours are preferred for intersections of boreholes with faults compared to areas with hot colours. In Fig. 7 the Malm $\delta$ − Purbeck unit is mostly blueish coloured which indicates the lowest slip tendency

values. Hence these are the best units for the intersection of boreholes with faults. An intersection with the northernmost fault in the red areas should be avoided.

## 6   Reliability of the model results

One of the key points in geomechanical modelling is the reliability of the model results in terms of the predicted processes and the presented multi-stage simulation of the in-situ stress field. As already mentioned in the result section 5 the calibration





procedure introduces uncertainties due to the low number of data points as well as their relatively large uncertainties. Further uncertainties are introduced by the model input, e.g. calibration data, rock properties, and structure. Hence the reliability of the model depends on the uncertainties of the input data used for the model. To quantify the model's reliability we use the already presented scalar value slip tendency (Morris et al., 1996) whose variability is introduced by the uncertainties in different input

data.

We compute the slip tendency for model scenarios which use the extreme values of the input parameters range of uncertainties. The model's linear elastic behaviour allows the individual quantification of the impact of different model parameter's uncertainties on the model's reliability. Therefore we compute several model scenarios in which sequentially only a single parameter is changed to an extreme value. This enables us to derive the individual impact of different parameters and hence

quantify the most important ones. The results of the slip tendency for each model scenario are subsequently compared to the "best-fit" slip tendency values from the "best-fit" model (Table 2). The variations of slip tendency introduced by the different independent parameters are added together which leads to an expected maximum variability in slip tendency of $\pm 0.57$.

The two main sources for the variability of slip tendency can be identified as the model-independent data for the $S_{Hmax}$ magnitude used for the model calibration and the rock properties density, Young's modulus, and Poisson ratio (Table 2). A

high variability of slip tendency of 0.14 is introduced by uncertainties in the $S_{Hmax}$ magnitude. Since the $S_{Hmax}$ magnitude is derived under several assumptions a wide range of possible $S_{Hmax}$ magnitudes is used for the calibration of the slip tendency model scenarios. Due to the fact that only limited knowledge and measurements of the rock properties are available a wide range of values are possible and they introduce a high variability of 0.18 in slip tendency.

Slip tendency proves to be quite robust ($\pm 0.01$) to the small uncertainties in the $S_{hmin}$ magnitude under the assumption that

the available data used for calibration is a valid proxy for the entire model (Table 2). Likewise only small variations in slip tendency are introduced by changes of $\pm 10°$ in the faults strike ($\pm 0.02$) and dip ($\pm 0.03$). The cohesion and friction angle act as more sensitive parameters (each $\pm 0.07$). Finally the two-stage calibration procedure itself introduces some moderate deviations ($\pm 0.05$) by the large amount and individual locations of calibration points used in the branch model.

## 7  Discussion

The objective of this work was to demonstrate the multi-stage approach for a high resolution 3D geomechanical numerical modelling workflow assessing the criticality in reservoirs. In contrast to a single model which includes both stress data records for calibration and high resolution representation of a local reservoir structure we use two models of different size. The regional scale root model is calibrated on stress data records and provides the stress state for the calibration of the reservoir scale branch model. This approach provides a cost efficient, quick, and reliable state-of-the-art calibration of geomechanical-numerical

models of the contemporary 3D in-situ stress field across scales. It is used to assess the criticality of reservoirs quantified by scalar values such as slip tendency. Furthermore, the approach provides the initial stress state for local application such as THM models.



## 7.1 Workflow

A single model with the same functionality as the two models in the multi-stage approach needs to account for the required high resolution in the reservoir area and the large model extent to include data for calibration. These two requirements are not contradictory per se but prolong the process of mesh-generation e.g. by the necessity to harmonise a regional scale low

resolution and a local scale high resolution structural model in the area of overlap. Furthermore, the manageability of the model (e.g. logical size) and the available time for computation (number of elements) in most instances requires a variable resolution which is refined only in the target are. Such a change in element size in a single model is possible but the mesh generation is cumbersome and needs a high number of elements. For a THM simulation of production and (re)injection incrementation over time significantly increases the computation time for each single element. Furthermore, in a single large model only a very

small area is of interest and hence large areas are simulated to no purpose while at the same time increasing the logical size, computation time, and effort.

If a multi-stage approach with two models is applied each model has its own fixed resolution with no required variations in element size (Fig. 2c). This significantly speeds up and simplifies the process of model generation since neither the structural models need to be harmonised nor a large difference in elements size needs to be implemented. Considering the same resolution

in the target area the time required for computation but even more the logical size of the models decreases improving the model's manageability. A geomechanical root model can also provide the stress state for a THM branch model which helps to save computation time by focussing the time-consuming THM simulation on the actual area of interest. Calibration data records for the additionally required scalar values on the pore pressure or temperature are provided in the literature or by dedicated models, e.g. Przybycin et al. (2015).

In addition the application of two models opens further possibilities to an improved and safer exploration and drilling. Structural features and stress magnitude measurements recorded during advanced exploration or even initial drillings can be implemented into the model workflow due to the simplified mesh re-generation. Even a change in the target area within the root model can be more easily implemented in the workflow since only a new branch model is required. The calibration of the root model can be updated with new stress data records as soon as they become available. Finally, a large calibrated root model

may include several target areas and can hence be reused and applied for more than one project.

## 7.2 Calibration

The two models in the presented two-stage approach are calibrated with different Dirichlet boundary conditions applied on an initial stress state. The root model is calibrated with data records which display the stress state as a result of the geologic history and tectonic evolution. The branch model, however, is calibrated on the stress state simulated in the root model. Both

calibration procedures are not limited in the number of calibration points and a weighting of the calibration points according to reliability can be easily realised. An extension of the two-stage approach to include three (or even more) models of different sizes is possible. Furthermore, the calibration procedure allows running several alternating models with different calibration data or differently weighted calibration points as well as variations in rock properties to quantify model specific variations. This



ability was used to quantify the reliability of the model's results. It is also useful for a future attempt to statistically determine uncertainties in the model's results.

Even without any additional computations a first order assessment of the impact of individual data records on the model calibration can be made by the assessment of changes in the boundary conditions. Therefore the "best-fit" boundary conditions

derived with and without certain data records are compared. Such a data record could be a newly performed hydraulic fracturing experiment which provides an additional $S_{hmin}$ magnitude data record. The variation of the derived boundary conditions induced by such a new data record provides a first idea of the variation of the stress state. Although, this feature cannot be used as a replacement for computations it helps to identify if the newly included calibration point yield a significantly different stress state which requires a reassessment of the situation or if the changes are minor and the exploration can be continued as planned.

The models showed in this work do not include any implicit faults. In theory, the calibration of a model including faults is possible as well. However, due to the non-linearities introduced by active faults the calibration process requires a regression analysis of a higher degree and hence several more test scenarios. This is beyond the scope of this work.

### 7.3   Model independent reliability

Apparently the model's reliability is mainly affected by the lack and high uncertainty of $S_{Hmax}$ magnitude data. The large

influence of the $S_{Hmax}$ magnitude is shown by two different models for viable $S_{Hmax}$ magnitudes in Fig. 6. A feasible method to narrow down the $S_{Hmax}$ variability is to enhance the knowledge of the Andersonian stress regime e.g. by gathering information on earthquake focal mechanism data (if available) or the crack orientation induced by leak-off tests or hydraulic fracturing (Haimson and Fairhurst, 1969; Hubbert and Willis, 1972; Zang and Stephansson, 2010). Such information is most likely available in the model area but not publicly accessible. Furthermore, an array of many expensive deep overcoring measurements

(several per borehole) could provide valuable information on the stress state and $S_{Hmax}$ in particular (Hast, 1969; Sjöberg et al., 2003).

The uncertainties related to the material properties are another large factor that limits the model's reliability. This can be mitigated at least partly by the usage of data from extensive databases (e.g. Bär et al., 2015; Lama and Vutukuri, 1978; Koch, 2009) or a by a conversion from seismic velocities which founds on empirical relations (Mavko et al., 2009). Finally, averaged

mean values from several laboratory tests of rock samples from the area and formation of interest is the safest but most expensive way to retrieve reliable information on the rock properties.

The uncertainties in the strike and dip of faults have a comparably small share in the reliability of the model while being challenging to mitigate due to the general uncertainties in the interpretation of 3D subsurface structures. The fault parameters cohesion and friction angle which are even more difficult to determine compared to the orientation reduce the model's reliability

to a slightly higher degree compared to strike and dip. The increase of the model's reliability by a better understanding of these parameters is possible but requires a detailed understanding of the great variability of in-situ fault zone behaviour and extent at depth.

Statistical methods to quantify uncertainties in the subsurface geometry exist for purely static structural models (Wellmann, 2013; Wellmann and Regenauer-Lieb, 2012). However, the computation time required extending this approach to 3D geome-





chanical numerical modelling approach and the ensuing analysis is beyond the scope of this work. A further investigation should be conducted as a sequel to the work by Bond et al. (2015) in a generic approach including geomechanical numerical modelling.

### 7.4 Model dependent reliability

Variations of the model results are also introduced by the multi-stage calibration approach itself and cannot be mitigated due to both models 3D stress state with lateral and vertical variations. The model's calibration, however, depends on the variations of only two independent boundary conditions. Additionally, small variations may be introduced by the model assumptions. However, these variations can be disregarded in the light of the major reasons for variations due to the small amount of stress magnitude data and rock properties. Table 2 clearly shows that any further advances in modelling are not efficient as long as

the amount and quality of input data ($S_{Hmax}$, rock properties) is not increased.

## 8   Application in geoengineering

Hydrocarbon reservoirs are currently exploited on a minor level in the Alpine Foreland (Lemcke, 1988; Sachsenhofer et al., 2006) and some of the former reservoirs are used for oil and gas storage (Sedlacek, 2009). However, hydrothermal reservoirs of economic interest for district heating or power generation are available (Lemcke, 1988; Bachmann et al., 1982; Fritzer et al.,

2012). These reservoirs are situated in highly karstified limestones of the Late Jurassic which are locally referred to as Malm formations (Lemcke, 1988). As of 2016 those deep reservoirs are already exploited by 21 municipal geothermal power plants and district heating projects of which Aschheim, Dürrnhaar, Erding, Freiham, Garching, Holzkirchen, Ismaning, Oberhaching, Poing, Riem, Sauerlach, and Unterschleißheim are in the root model area (Bundesverband Geothermie, 2016). Borehole data from these projects could be easily implemented in the calibration of the root model and would increase its reliability if they

would become publicly available.

Within the root model perimeter several geothermal projects are currently in planning stage which are namely Bernried, Gräfelfing/Planegg, Königsdorf, Markt Schwaben, Puchheim/Germering, Raststätte Höhenrain, Starnberg, Weilheim/Wielenbach, and Wolfratshausen (Bundesverband Geothermie, 2016). In addition the municipal energy supplier of Munich (SWM) plans to install an extensive geothermally driven district heating grid for the entire city (Stadtwerke München GmbH, 2012). Therefore

a 3D seismic survey was conducted in the entire southern parts of Munich in winter 2015/16 (Bundesverband Geothermie, 2015). The presented root model provides data for a first order assessment of the in-situ stress state at the exact locations of these planned geothermal projects. Furthermore it provides calibration data for local and/or reservoir scale models based on high resolution 3D seismic surveys which simulate the stress state, its criticality, and the possibility of subsidence due to the production and reinjection of fluid and heat.

Furthermore, the two-stage approach could be extended to a three-stage approach which incorporates a global model of the entire Bavarian Molasse Basin. More data for calibration as well as more potential applications might be available in such an




enhanced area. Thereby the regional or global root model could be established as a *community model* which provides the stress state for further applications and/or local models for planned projects.

## 9 Conclusions

In this work we present a multi-stage 3D geomechanical numerical modelling approach which provides a cost efficient, reliable, and fast procedure to generate and evaluate the criticality of the stress state in a small target area where in general no stress data for model calibration are available. The approach uses a large scale root model which is calibrated on available stress data and a small scale branch model which is calibrated on the root model. We exemplify this in a two-stage approach in the German Molasse Basin around the municipality of Munich. The discussion of reliability of the model results clearly shows (1) that variations are large and (2) that they are mainly introduced by the uncertain material properties and missing $S_{Hmax}$ magnitude data. At this stage, the model's quality depends on the amount and quality of available input data and not on the modelling technique itself. Any further improvements in the model's resolution and applied techniques will not lead to an increase in reliability. This can only be achieved by more and high-quality data for calibration.

*Acknowledgements.* The research leading to these results has received funding from the European Community's Seventh Framework Programme under grant agreement No. 608553 (Project IMAGE). The authors would like to thank Thomas Fritzer (LFU Augsburg) for his support, Dietrich Stromeyer for discussing the procedure and the programming of the stress data analysis tool, and Arno Zang for his comments which significantly improved the manuscript. The map is prepared with the Generic Mapping Tool GMT (Wessel et al., 2013) using SRTM topographic data (Farr et al., 2007).
The service charges for this open access publication have been covered by a Research Centre of the Helmholtz Association.



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



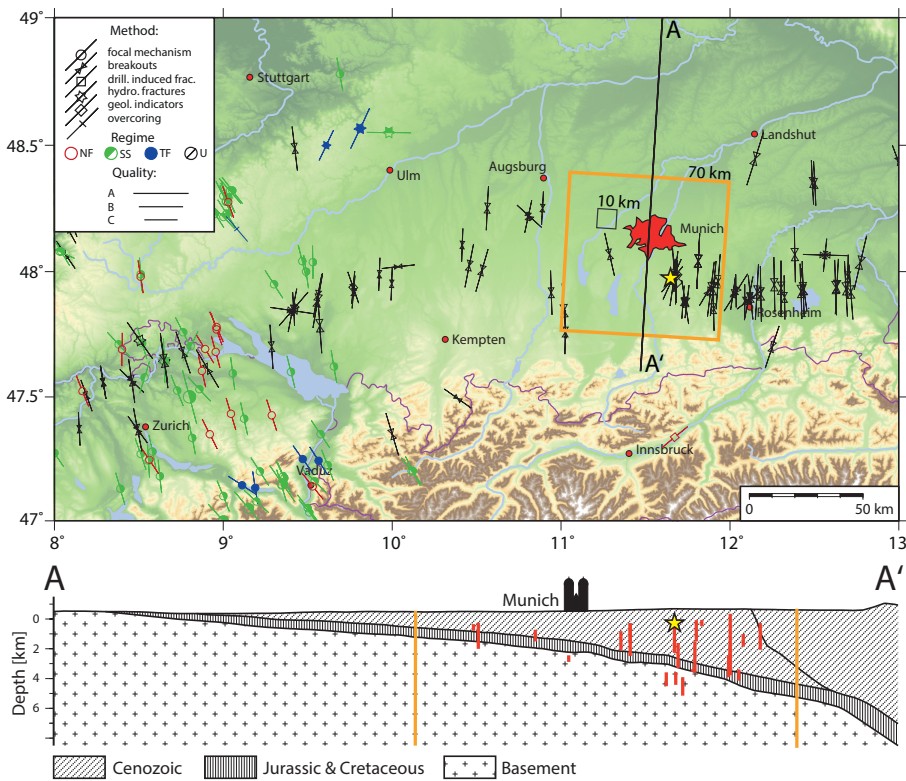

**Figure 1.** Stress Map of the Bavarian Molasse with 172 A-C quality data records based on the World Stress Map database release 2008 (Heidbach et al., 2008, 2010) and additional data from Reiter et al. (2015) and Heidbach and Reinecker (2013). Lines show the $S_{Hmax}$ orientation with line length proportional to WSM data quality (Heidbach et al., 2010). Colour code of the data shows the stress regime with red for normal faulting (NF), green for strike-slip (SS), blue for thrust faulting (TF), and black if the regime is unknown (U). The star marks the location of the Sauerlach project where information on the $S_{hmin}$ magnitude is available (Seithel et al., 2015). The orange box shows the lateral boundaries of the 3D geomechanical-numerical model area ($70 \times 70 \text{km}^2$) and the small black box indicates the typical dimensions of a reservoir model ($10 \times 10 \text{km}^2$). The cross section A-A' (based on Przybycin 2015) shows a 1:2.5 exaggeration of the area with red lines being the borehole sections with stress indicators within the model area.




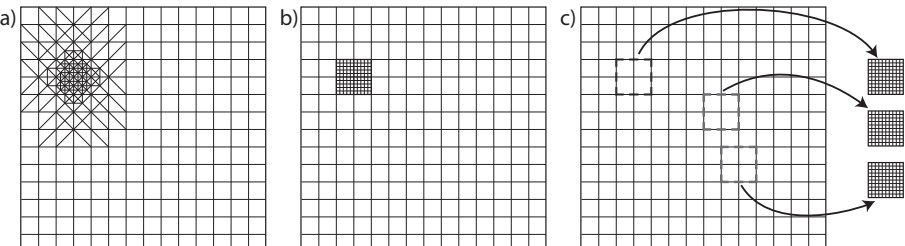

**Figure 2.** Different types of modelling approaches: a) A refined mesh in the area of interest is expensive and inefficient due to the large amount of elements required for the discretisation of the gradient in resolution. Furthermore, it requires a complete re-mesh and re-evaluation in case of any changes in the geometry or input data. b) A local model nested within a regional model matches the variables at the boundary. A complete re-mesh and re-evaluation is required in case of geometry or input data changes. c) A multi-stage approach has the easiest mesh generation since the differently sized models are generated independently. Furthermore, several reservoir models can be based on the same regional model.

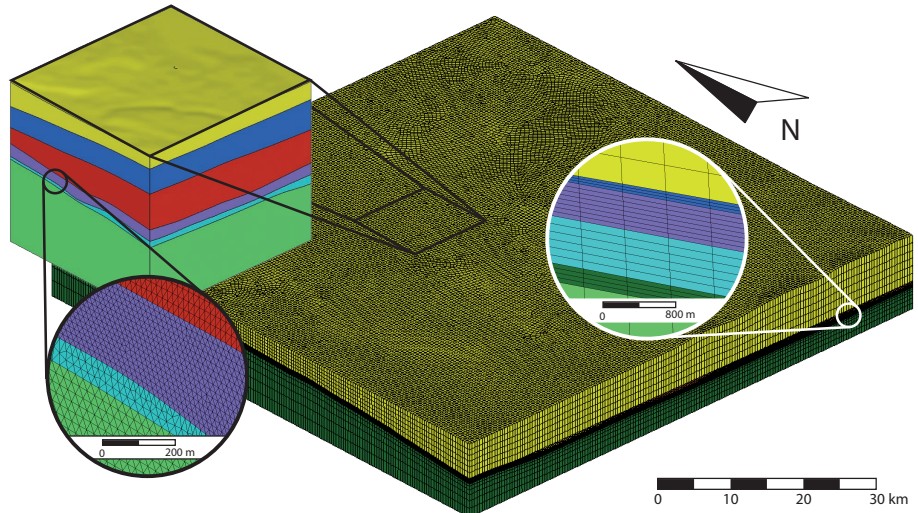

**Figure 3.** The root and branch model discretised with $10^6$ hexahedral and $21 \times 10^6$ tetrahedral elements respectively. Please note that to improve visibility the discretisation of the branch model is only displayed within the magnified inset. Both magnified regions show the Malm $\alpha-\gamma$ (turquois) and Malm $\delta-\zeta$ and Purbeck (purple) units which are the predominant target units for geothermal exploration in the Bavarian Molasse Basin.




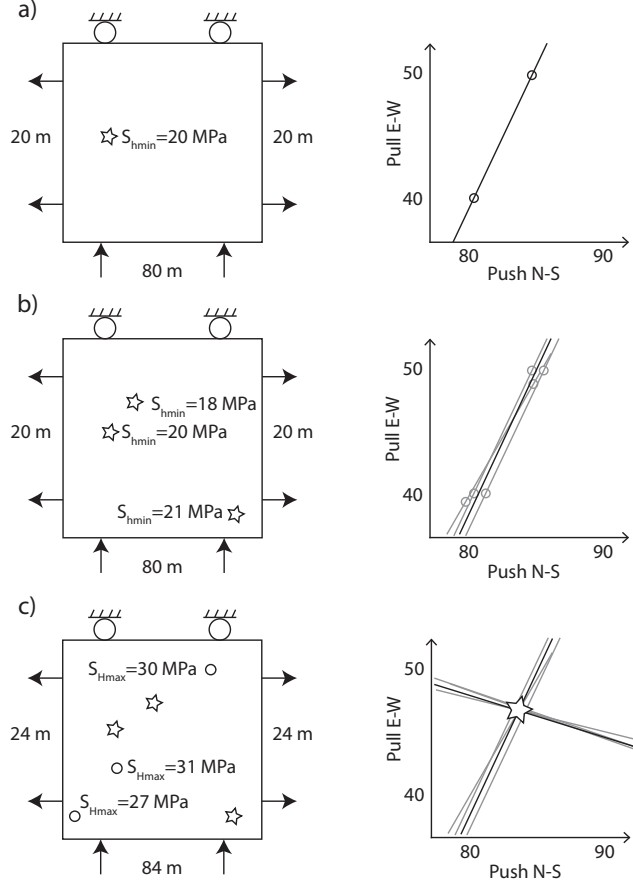

**Figure 4.** Left: Exemplified schematic models with the data records used for calibration (star: $S_{hmin}$ magnitude, circle: $S_{Hmax}$ magnitude). Right: Linear slopes that display the magnitudes of possible combinations of displacement boundary conditions applied normal to the the E-W (y-axis) and N-S (x-axis) sidewalls of the model. For each data record an individual slope defines the possible combinations of boundary conditions to fit the model to this calibration data record. a) A single $S_{hmin}$ magnitude can be calibrated by an unlimited amount of combinations of boundary conditions which are on a linear slope. b) Several $S_{hmin}$ magnitudes usually cannot be calibrated to an exact fit. However, a linear regression of all the linear slopes derived for the calibration of each individual data record provides a "best-fit" slope. This slope defines combinations of "best-fit" boundary conditions that fit the data records used for calibration equally well. c) Several $S_{hmin}$ and $S_{Hmax}$ magnitude data records used for calibration result for each $S_{hmin}$ and $S_{Hmax}$ in a linear slope of combinations of "best-fit" boundary conditions. At the intersection of these two slopes the "best-fit" boundary conditions (indicated by a star) are found for the calibration of both $S_{Hmax}$ and $S_{hmin}$ together.





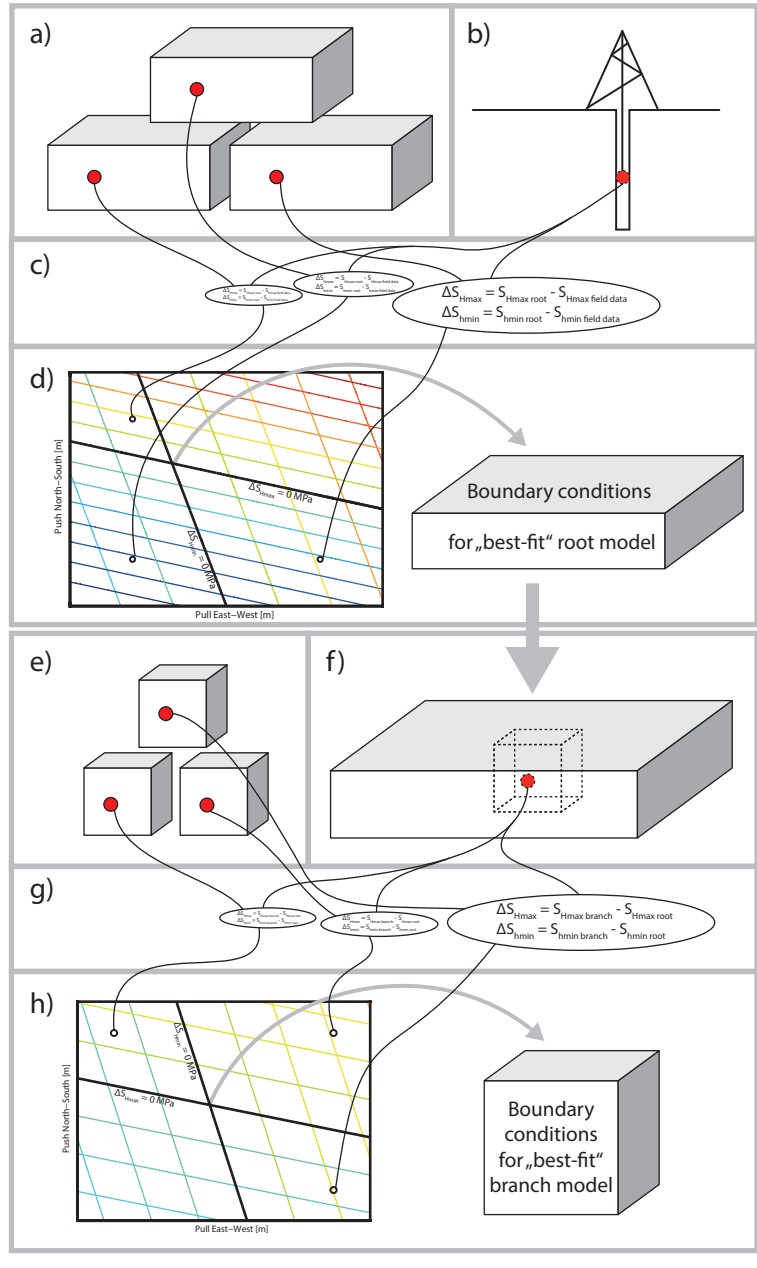

**Figure 5.** The calibration workflow for the root and branch model. (a) Three models with different Dirichlet boundary conditions provide stress data comparison values for a calibration with (b) observed magnitude stress data. The deviation of the modelled from the observed stress state of each of the three scenarios (c) is used in a linear regression to derive the boundary conditions to compute the "best-fit" root model (d). (e) Three different branch models provide stress data comparison values for a calibration with magnitude data from the "best-fit" root model (f). The deviation of the modelled stress state to that provided from the root model for each of the three scenarios (g) is used in a linear regression to derive the boundary conditions required to compute the "best-fit" branch model (h).



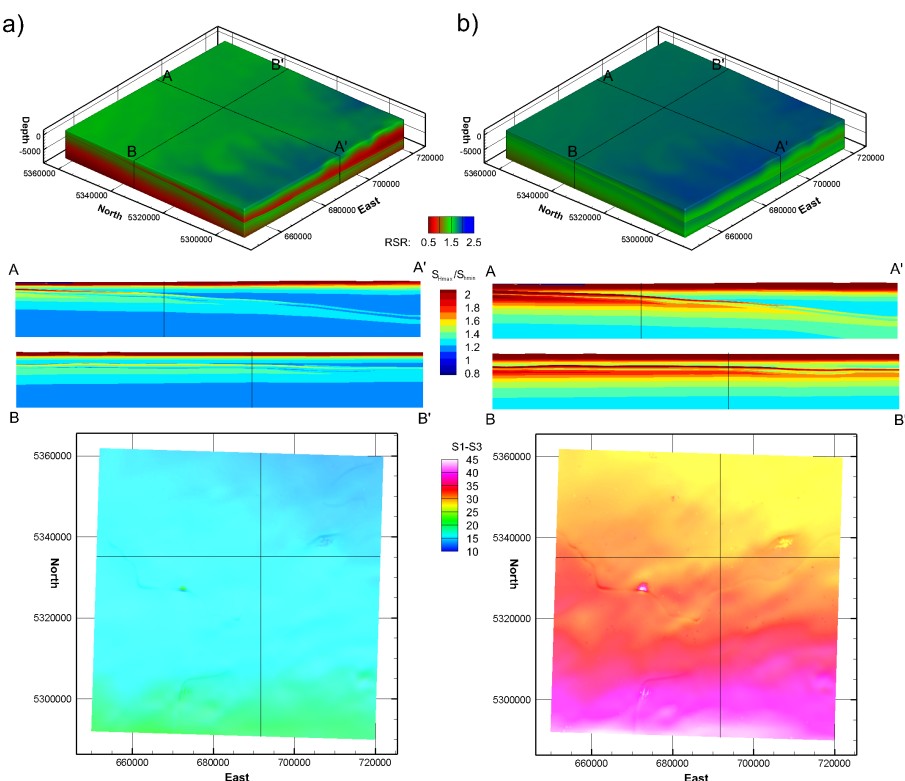

**Figure 6.** Results of the "best-fit" root model (b) and an alternative scenario that fits the $S_{hmin}$ values equally well, but is calibrated against a lower $S_{Hmax}$ value (a). The overall difference that results from the different $S_{Hmax}$ values used for the calibration is expressed in the continuous scale of the stress regime ratio (RSR) which is between 0.5 (normal faulting regime), 1.5 (strike slip), and 2.5 (thrust faulting regime) (Simpson, 1997) in the model volume. The horizontal stress anisotropy expressed in the ratio of $S_{Hmax}/S_{hmin}$ is shown on two cross sections which intersect below Munich. The differential stress (difference between the maximum and minimum principal stress, lowermost panel) is mapped on a surface which is vertically centered in the Malm $\alpha - \gamma$ units.





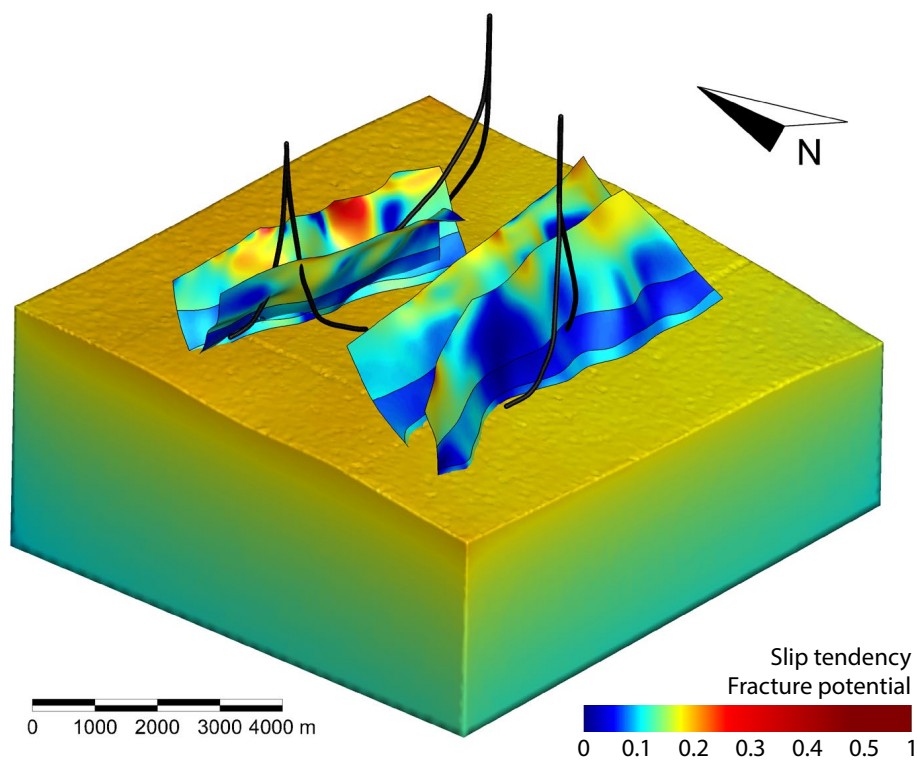

**Figure 7.** The generic branch model results are shown by means of slip tendency (ST) values (Morris et al., 1996) mapped on generic faults in the Chattian, Purbeck, and Malm units and by means of the Fracture Potential (FP) (Connolly and Cosgrove, 1999) displayed for the model volume of the basement. Both values vary from zero to one indicating low and high criticality, respectively. Note, that the colour map of these values is non-linear. The results clearly indicate that the generic faults are far away from failure with the largest value of ST of 0.3. The low FP values (max. 0.38) e.g. give an estimate on how much fluid pressure would be needed to fracture the intact rock in a stimulation experiment to enhance the permeability.





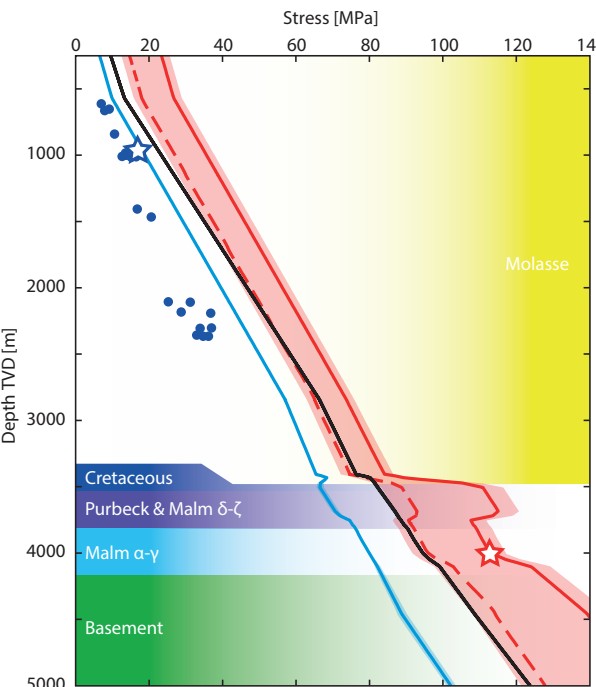

**Figure 8.** Stratigraphy and model result of the root model along the borehole of the geothermal project in Sauerlach. Lines show the results of the "best-fit" root model: blue for the $S_{hmin}$ magnitude, black for the vertical stress and red for the $S_{Hmax}$ magnitude. The blue dots are formation integrity tests (FITs) which are a lower boundary for the magnitude of $S_{hmin}$ and not used for calibration, the blue star represents the suspected LOT, the red star shows the $S_{Hmax}$ magnitude of 112MPa used for calibration (Seithel et al., 2015). Shaded areas in the same colour around the lines show the range of model scenarios that fit equally well to the model-independent constraints. The dotted red line shows the $S_{Hmax}$ magnitude for the model scenario in Fig. 6a.



**Table 1.** The stratigraphic units, their discretisation, and according rock properties which are present in the root and branch models. Units which are only preserved in parts of the root model are marked with *.

| Unit(s) | Root model: vertical layers | Branch model: number of elements | Density [kg m$^{-3}$] | E-module [GPa] | Poisson ratio |
|---------|------|------|------|------|------|
| Molasse | 6 | - | 2375 a,b | 15 c | 0.29 c |
| Upper Molasse | - | $1.1 \times 10^6$ | 2375 a,b | 15 c | 0.29 c |
| Aquitanian | - | $2.3 \times 10^6$ | 2495 d | 32.5 d | 0.21 d |
| Chattian | - | $7.6 \times 10^6$ | 2758 e | 39 d | 0.23 d |
| Cretaceous | 3* | - | 2647 a,b | 22.5 b | 0.25 b |
| Malm $\delta$-Purbeck | 8 | $6.3 \times 10^6$ | 2667 d,e | 40 b | 0.25 b |
| Malm $\alpha - \gamma$ | 6 | $2.2 \times 10^6$ | 2460 d | 30 b,c | 0.29 b,c |
| Pre Malm | 4* | - | 2680 a,b | 20 c | 0.25 c |
| Crust | 6 | $2.2 \times 10^6$ | 2850 a | 45 c | 0.24 c |

a: Przybycin (2015), b: Koch (2009), c: Hergert et al. (2015), d: Lama and Vutukuri (1978), e: Koch and Clauser (2006)

**Table 2.** The expected maximum variations in slip tendency (ST) introduced by the uncertainties of the model parameters. This comparison is made at 40 locations in the Malm $\alpha - \gamma$ and Purbeck target units and an arithmetic mean is computed for each model parameter.

| Source of uncertainty | | $\Delta_{max}$ ST |
|---|---|---|
| Rock properties | | 0.18 |
| Calibration | $S_{Hmax}$ | 0.14 |
| | $S_{hmin}$ | 0.01 |
| Analysis | Strike $\pm 10°$ | 0.02 |
| | Dip $\pm 10°$ | 0.03 |
| | Cohesion $\pm 5$MPa | 0.07 |
| | Friction angle $\pm 10°$ | 0.07 |
| Two-stage calibration | | 0.05 |
| Total variations | | 0.57 |