# Peer review of "A multi-stage 3D stress field modelling approach exemplified in the Bavarian Molasse Basin"

_Solid Earth, 2016_

## Short Comment (SC1) · 19 Jul 2016

The paper presents first a method for modelling the regional stress field as determined from data gathered in the world stress map. It is used then to evaluate the criticality of the loading of faults susceptible to be reactivated by human exploitation of local fluids in domains where no local stress measurement is available. The objective is to take advantage of stress measurements to define first a regional stress field that fits observations and then to use the model, through a locally refined mesh, for evaluating the criticality of the faults of concern. The regional model concerns a 70 km x 70 km x 10 km volume located in the vicinity of Munich, in the Bavarian Molasse Basin. It assumes an elastic behavior for the various geomaterials involved and a classical friction law for the stability of faults. This modelling procedure is quite classic and rests on the validity of a few basic hypotheses that need to be better discussed. Indeed, authors

assume that vertical direction is a principal direction throughout the modelled volume. But a recent paper by Maury et al. (Geophysical Journal Int., 2014, vol. 199, pp 1006-1017) have suggested that in this area the stress field at depth is controlled by the fossil Alpine subduction. More precisely the steeply dipping Lithosphere-Asthenosphere contact encountered around 50 to 70 km in this area supports only a pressure so that along this contact none of the principal directions are vertical. This applies to a depth of the same order of magnitude as the horizontal extensions of the domain of interest. In addition, authors should better document the topography of the sediment-basement contact at the 70 km x 70 km scale, for it is likely not horizontal. In other words, in order to be credible, the model should extend much deeper than 10 km, given its 70 x 70 km horizontal extent. Another important issue concerns the validity of an elastic hypothesis for modelling the present day stress field. Indeed, recent GPS measurements (Nocquet, 2012, Tectonophysics, vol. 579, pp 220-242) show no present day measurable displacement in this area so that the displacement considered by authors as boundary conditions are likely to be associated with the Alpine tectonics. But this tectonic activity ceased being active some 4 to 5 million years ago. It is most likely, as suggested by the discussion by Gunzburger and Cornet, 2007, (referenced in text) that the equivalent elastic parameters used for modeling the long term deformation of such visco-elastic materials are much softer than those described in table 1. This needs a detailed discussion. Also of import are the stress discontinuities observed at the limits between the various geomaterials. Is there no limit to the maximum "stress jump" described on figure 8 ? Finally, the classical proposition that the criticality of faults is well described by a Coulomb type failure mechanism requires also a better discussion. Indeed, some not so recent work (Sulem et al., 2004, C.R. Geosciences, vol. 336, pp 455-466, Sulem, 2007; Tectonophysics vol. 442, pp 3-13) suggests that the mechanical behavior of faults is not properly represented by a Coulomb failure criterion. The role of long term stress relaxation in the gauge material should be discussed. In summary, the model described in this paper follows classically accepted hypotheses. But "not so recent" results on the role of geomaterial rheology have shed some doubts on

these hypotheses and a modern publication cannot ignore these developments. Only after these various points have been discussed properly, may I recommend publication of this paper.

Francois Cornet, reviewer

---

## Referee Comment (RC2) · Anonymous Referee #2 · 1 Aug 2016

This is a very nice contribution that brings a new methodology of extrapolating and understanding the stress field distribution obtained from widely distributed and unevenly spaced data towards the resolution required by the exploration industry. Although the individual components rely mostly on known methodological approaches, their combination is indeed novel and highly interesting. I agree with the authors that such an approach has multiple applications, in particular for geoengineering and geothermal exploration. I see only a few concerns on the general approach of such a methodology that can, potentially, be better discussed in the manuscript: (1) the resolution of the input geological model is given as granted and no feedback between the modelling inferences and the distribution of stratigraphy and faults geometry. Such a modelling approach should contain feedbacks to known fault kinematic behaviour that may correct and improve the reliability of the modelled predictions. Inspecting the overall

geological input model shows that the resolution is coarser that the density of even the root model in many areas. I advise the authors to discuss more the role of the local geological distribution of faults into the modelling results and the rather underestimated impact of (strain) partitioning along the large structural lineaments. This is quite vaguely discussed. I furthermore agree with previous reviewers that the investigation depth is somehow limited given the much deeper extent of the overall process driving the present-day stress distribution; (2) the overall world stress data work very well to regional estimates of the state of stress, but their reliability significantly decreases at higher resolution due to partitioning and local distribution effects. Although the stress data distribution appear simple in the study area, it would be good to have a discussion in a resolution analysis applied to the modelling results; (3) the elastic approach considered is somehow limited given the wide diversity of observed scenarios for instance controlling strain weakening and strain hardening in fault (re)activations, generally derived by experimental studies and tested by observations, e.g. in areas affected by induced or triggered seismicity. It would be good to have a better discussion of the link between the model and such a variability of deformation mechanics.

---

## Author Comment (AC1) · 1 Sep 2016

Reply to reviewer #1

| Reviewer's Comment | Author's reply | Action |
|---|---|---|
| Authors assume that vertical direction is a principal direction throughout the modelled volume. But a recent paper by Maury et al. (2014) have suggested that in this area the stress field at depth is controlled by the fossil Alpine subduction. More precisely the steeply dipping Lithosphere-Asthenosphere contact encountered around 50 to 70 km in this area supports only a pressure so that along this contact none of the principal directions are vertical. | We agree with the reviewer that the vertical direction cannot be assumed as a principal stress axes everywhere. We do not assume this in the model. We only make this assumption when talking about the reduced stress tensor (Zoback, 2010) in conjunction with collecting data for calibration. We added a sentence to highlight this. | "Only the orientation of the reduced stress tensor and to a lesser extent information on the stress regime are relatively good estimated from stress indicators." p3, l7-8 |
| Authors should better document the topography of the sediment-basement contact at the 70 km x 70 km scale, for it is likely not horizontal. In other words, in order to be credible, the model should extend much deeper than 10 km, given its 70 x 70 km horizontal extent. | We acknowledge that our wording might be not detailed enough here. Indeed, the topography of the sediment-basement contact is not horizontal but a surface just like any other surface between geological bodies in the model. It is documented in detail in the model published by Przybycin (2015, referenced in the manuscript). The bottom of the model however is a horizontal surface which is entirely composed of basement/Upper Crust rock. We modified our wording accordingly. In contrary to the mentioned modelling approach by Maury et al. (2014) our presented modelling approach deals with the stress state in the upper crust. It has been shown by Reiter & Heidbach (2014) and Hergert et al. (2011) that the geometry of the Moho and other very deep structures only play a very minor part in the modelling of the stress state of the upper crust. Especially in light of the | - "The part of the structural model used for the geomechanical model has a size of 70x70 km² and is referred to as the root model. It includes the sediments in the Molasse Basin in their entire vertical extent. The bottom of the model is situated at a depth of 9 km entirely within the Upper Crust." p4, l17ff
- In several instances we emphasized that the model is only for the upper part of the crust.
- We discuss the influence of deep processes in section 7.4 model dependent reliability |

| | large uncertainties mainly due to the SHmax magnitude and material properties it is justified to concentrate on those larger uncertainties. | |
|---|---|---|
| Another important issue concerns the validity of an elastic hypothesis for modelling the present day stress field. Indeed, recent GPS measurements show no present day measurable displacement in this area so that the displacement considered by authors as boundary conditions are likely to be associated with the Alpine tectonics… | We agree with the reviewer that the application of displacement boundary conditions derived from a measured displacement are in this case not a valid method to calibrate the model. Again our wording was not detailed here and we modified it accordingly.
We only use displacement boundary conditions to initiate the stress field. We do not calibrate the model on the prescribed displacement but on the stresses which are modelled by the application of displacement boundary conditions. In other words, we do not place any significant meaning on the amount of displacement applied to the model. | "Dirichlet boundary conditions (i.e. displacements) are applied to the sidewalls of the model to create horizontal differential stresses. The boundary conditions are adjusted in a way that the modelled magnitude of $S_{Hmax}$ and $S_{hmin}$ at the calibration points fit the observed magnitudes." p6, l23ff |
| Also of import are the stress discontinuities observed at the limits between the various geomaterials. Is there no limit to the maximum "stress jump" described on figure 8 ? | The "stress jumps" which are observed at the contacts between different geomaterials are regularly observed in situations where two materials of very different elastic properties are in contact to each other. In the real world these jumps are possibly smoother since the associated contact zone has evolved with time and are hence not as "jumpy" and sudden as in the model.
Such a smoother transition is possible to realise in a model. However, the limited and missing knowledge of the actual contact behaviour at depth shows that such an approach is not beneficial because the uncertainties would increase dramatically. | - |

| | | |
|---|---|---|
| Finally, the classical proposition that the criticality of faults is well described by a Coulomb type failure mechanism requires also a better discussion. Indeed, some not so recent work suggests that the mechanical behavior of faults is not properly represented by a Coulomb failure criterion. The role of long term stress relaxation in the gauge material should be discussed. | We agree with the reviewer that more accurate failure criteria than Mohr-Coulomb do exist. Such more elaborate criteria can also be applied to analyse our model results. However, as for example shown by Sulem (2007) more accurate failure criteria are dependent on high quality information of the rock material. We do not have access to such data for the according materials and thus the uncertainties would be very high when assuming standard values. Hence in this example we remain with the more basic but still frequently applied Mohr-Coulomb criteria. That does by no means imply that our presented approach does not support the application of more elaborate failure criteria. On the contrary the model results can be analysed with all kinds of failure criteria. However, in the lights of the already high uncertainties we refrained from adding even more uncertainties by the application of a failure criterion which is very exact for a specific rock but might not be applicable for the material in our model. We added a sentence to explain this issue. | "It is used to assess the criticality of reservoirs which can be quantified by scalar values such as slip tendency. If detailed information on the fracture behaviour of the rock are known more elaborate fracture criteria than Mohr-Coulomb (e.g. Sulem2007, Zang2010 )can be applied to analyse the model results." p11, l33ff |

---

## Author Comment (AC2) · 1 Sep 2016

Reply to reviewer #2

| Reviewer's Comment | Author's reply | Action |
|---|---|---|
| (1) the resolution of the input geological model is given as granted and no feedback between the modelling inferences and the distribution of stratigraphy and faults geometry. Such a modelling approach should contain feedbacks to known fault kinematic behaviour that may correct and improve the reliability of the modelled predictions. Inspecting the overall geological input model shows that the resolution is coarser that the density of even the root model in many areas. I advise the authors to discuss more the role of the local geological distribution of faults into the modelling results and the rather underestimated impact of (strain) partitioning along the large structural lineaments. This is quite vaguely discussed. I furthermore agree with previous reviewers that the investigation depth is somehow limited given the much deeper extent of the overall process driving the present-day stress distribution; | - We agree with the reviewer on the importance to highlight that our approach is able to handle the inclusion of faults and modified the text accordingly.
- The finer resolution compared to the input model is a result of the desire to have smooth surfaces to supress stress peaks at artificial element boundaries. This results in a slightly finer resolution. We improved our wording here.
- As pointed out in the reply to reviewer #1 the maximum depth of the model does not mean that the important processes are disregarded. However, the modelling approach founds on the calibration of the static stress state on observed stress data records and not on the derivation of the stress state by the simulation of processes. Hence the underlying deep-seated processes are not simulated but their resulting effect on the stress state is included in the model since the stress data used for calibration is a result of these processes. This approach is beneficial due to the shallow area of interest. We altered the wording to be more precise. | - "The models showed in this work do not include any implicit faults and no strain partitioning is assumed." p13, l17ff.
- p4, l14ff
- "Dirichlet boundary conditions (i.e. displacements) are applied to the sidewalls of the model to create horizontal differential stresses. The boundary conditions are adjusted in a way that the modelled magnitude of SHmax and Shmin at the calibration points fit the observed magnitudes." p6, l23ff |

| | | |
|---|---|---|
| (2) the overall world stress data work very well to regional estimates of the state of stress, but their reliability significantly decreases at higher resolution due to partitioning and local distribution effects. Although the stress data distribution appear simple in the study area, it would be good to have a discussion in a resolution analysis applied to the modelling results; | We agree with the reviewer and we added some lines on local stress perturbations in the discussion section. Furthermore we highlight the importance of representative calibration data. | "In the presented region the stress field is very homogeneous but in other regions significant local lateral variations exist and need to be accounted for. This can be accomplished for example by lateral variations of the material properties or faults. It is crucial to ensure that the data used for the calibration is representative for the regional material and geometry in the root model." p12, l32ff |
| (3) the elastic approach considered is somehow limited given the wide diversity of observed scenarios for instance controlling strain weakening and strain hardening in fault (re)activations, generally derived by experimental studies and tested by observations, e.g. in areas affected by induced or triggered seismicity. It would be good to have a better discussion of the link between the model and such a variability of deformation mechanics. | We agree with the reviewer that every model is somehow limited in the amount of processes which are represented. In this situation we consider an elastic approach to be sufficient especially in light of the high uncertainties of the input parameters. Anyway, the processes which are considered are highly dependent on the situation described by the model. We highlighted this in the discussion. | "The inclusion of faults makes sense in situations where detailed information on fault geometry, extent, and parameters are available and a significant impact of the faults on the regional stress field or a (re)activation is expected. However, in this example, the available stress data suggests that no faults with a major impact are located within neither the root model nor the branch model area. The calibration of a model including faults and fault specific behaviour, e.g. strain weakening or hardening or long-term relaxation of the gauge material, is possible as well." p14, l17ff |

---

## Author Comment (AC3) · 1 Sep 2016

For the author's reply to this short comment please refer to our answer to RC1 (reviewer comment 1). The comments of RC1 and SC1 are identical.